# Multiple Physiological and Biochemical Functions of Ascorbic Acid in Plant Growth, Development, and Abiotic Stress Response

**DOI:** 10.3390/ijms25031832

**Published:** 2024-02-02

**Authors:** Peiwen Wu, Bowen Li, Ye Liu, Zheng Bian, Jiaxin Xiong, Yunxiang Wang, Benzhong Zhu

**Affiliations:** 1College of Food Science & Nutritional Engineering, China Agricultural University, Beijing 100083, China; wpw1026@126.com (P.W.); b20223060519@cau.edu.cn (B.L.); liuliuye7@163.com (Y.L.); bianzheng0311@163.com (Z.B.); sapphire_x7@163.com (J.X.); 2Institute of Agri-Food Processing and Nutrition, Beijing Academy of Agricultural and Forestry Sciences, Beijing 100097, China

**Keywords:** ascorbic acid, cofactor, substrate, antioxidant, pro-oxidant, abiotic stress

## Abstract

Ascorbic acid (AsA) is an important nutrient for human health and disease cures, and it is also a crucial indicator for the quality of fruit and vegetables. As a reductant, AsA plays a pivotal role in maintaining the intracellular redox balance throughout all the stages of plant growth and development, fruit ripening, and abiotic stress responses. In recent years, the de novo synthesis and regulation at the transcriptional level and post-transcriptional level of AsA in plants have been studied relatively thoroughly. However, a comprehensive and systematic summary about AsA-involved biochemical pathways, as well as AsA’s physiological functions in plants, is still lacking. In this review, we summarize and discuss the multiple physiological and biochemical functions of AsA in plants, including its involvement as a cofactor, substrate, antioxidant, and pro-oxidant. This review will help to facilitate a better understanding of the multiple functions of AsA in plant cells, as well as provide information on how to utilize AsA more efficiently by using modern molecular biology methods.

## 1. Introduction

Ascorbic acid (AsA, also known as vitamin C) is an essential nutrient and one of the most frequently used supplements in humans due to our lack of L-gulono-1,4-lactone oxidase (GuLO), a key enzyme contributing to the conversion of L-gulono-1,4-lactone to AsA [1,2,3]. Although AsA has four optical isomers in its natural state, including D-ascorbic acid, D-isoascorbic acid, L-ascorbic acid, and L-isoascorbic acid [4] (Figure 1A), L-ascorbic acid has the highest bioactivity as well as the highest abundance in plant cells, while the remaining three forms have no or only a little bioactivity [4]. Therefore, AsA in this review especially refers to L-ascorbic acid, which is a water-soluble cyclic lactone with a similar structure to glucose. An aqueous solution of AsA is acidic, after the enol hydroxyl group at the second and third carbon atoms (C2 and C3) dissociates hydrogen ions (H^+^) [4]. More importantly, the provision of electrons in the enol hydroxyl group provides the foundation for AsA as an important intracellular reductant for plants [5].

In order to regain electrons, the interconversion of AsA and dehydroascorbic acid (DHA) in plant cells is required, which is the so-called ASA-GSH cycle [6] (Figure 1B). This process begins with the oxidation of AsA by ascorbate peroxidase (APX) to produce monodehydroascorbic acid (MDHA), which is unstable in cells and prone to spontaneous disproportionation to generate DHA [6]. Both MDHA and DHA are reduced to AsA, which is accomplished by monodehydroascorbate reductase (MDAR) and dehydroascorbate reductase (DHAR), respectively [6]. The reduction process of DHA to AsA is accompanied by the mutual conversion between glutathione (GSH) and oxidized L-glutathione (GSSG) [6]. Therefore, the AsA-GSH cycle is the foundation of providing electrons for many important physiological and biochemical pathways to maintain the redox balance and respond to abiotic stresses in plants [6].

At present, the plant AsA biosynthesis pathway, as well as its transcriptional and post-transcriptional regulation, have been clarified [2] (Figure A1). However, a comprehensive and systematic summary of AsA-involved cellular biochemical pathways, as well as AsA’s corresponding physiological functions in plants, is still lacking. Here, we review the multiple roles of AsA, including its involvement as a cofactor, substrate, and antioxidant, which is helpful for utilizing AsA more efficiently using modern molecular biology methods.

## 2. As a Cofactor, AsA Maintains the Active Center of the 2-Oxoglutarate/Fe (II)-Dependent Dioxygenases

The reversible oxidation of AsA plays an indispensable bioactive role in maintaining the active center of the 2-oxoglutarate/Fe (II)-dependent dioxygenase (2-ODD), a large-class protein family in plants that participates in hydroxylation, desaturation, and demethylation throughout the entire stages of plant growth, development, fruit ripening, and abiotic stress response [7,8,9] (Table 1). These reactions are required to control proteostasis and regulate chromatin accessibility, which explains AsA’s dominance in cellar homeostasis [8]. For example, it is involved in the control of plant hormones such as gibberellins (GA), salicylic acid (SA), ethylene, and indoleacetic acid (IAA) [10]. The 2-ODDs are also involved in the synthesis of various secondary compounds, such as flavonoids and glucosinolates, suggesting that AsA is required for fruit nutrient quality [11,12]. Furthermore, the assembly of the cell wall and hypoxia response requires hydroxylation catalyzed by 2-ODDs, demonstrating the role of AsA in maintaining plant cell rigidity [13]. Canonical structural features of 2-ODDs have been revealed, including a double-stranded β-helix core fold known as the jellyroll topology, which supports and protects catalytic triads of the Fe (II)-binding sequence [14]. The highly conserved but not widely distributed HX(D/E)XnH triplet motif that is present in this sequence is essential for Fe (II) binding [8,15,16]. The catalytic process of 2-ODDs is as follows [15]: (1) 2-oxoglutarate binds to a 2-ODD’s substrate, replacing H_2_O from Fe (II). (2) Molecular O_2_ binds to Fe (II), catalyzing 2-oxoglutarate decarboxylation to produce succinate. (3) The active Fe (II) center of the 2-ODD’s substrate is oxidized to Fe (III), which is subsequently reverted to Fe (II), accompanied by the release of the substrates and CO_2_. It is generally believed that the mechanism of AsA functioning in the above process is the reduction of Fe (III) to Fe (II) for the 2-ODD’s catalyzed center during plant growth and development [17].

### 2.1. AsA Is Necessary for Flavonoid Synthesis

AsA is required as a cofactor in the flavonoid synthesis pathway, by four members of the 2-ODDs superfamily: flavonoid 3β-hydroxylase (F3H), flavonoid synthase I (FNS I), flavonoids synthase (FLS), and anthocyanin synthase (ANS)/white anthocyanin dioxygenase (LDOX) [11] (Figure 2). In the flavonoid pathway, these four 2-ODDs enzymes perform their distinct roles by catalyzing the hydroxylation and desaturation of three different rings on the 2-phenyl chromene framework, respectively, to produce various flavonoid metabolites [21]. According to the chemical structures, flavonoids have been classified into six subclasses based on their chemical structures: flavan-3-ols, flavanones, flavones, flavonols, isoflavones, and anthocyanins [22].

Dihydroflavonols are produced when F3H introduces a hydroxyl group to the C3 position of (2S)-flavonones [11,21]. For F3H to exert enzymatic activity, it must combine with Fe (II), 2-oxoglutarate and AsA; when these three cofactors are present alone, F3H is inactive [23,24,25]. This assay for enzymatic activity in vitro indicates that 67 μM AsA has the remarkable effect of protecting the recombinant *Petunia hybrida* F3H diethylpyrocarbonate-induced inactivation, thereby restoring the activity to 93% as compared to the normal conditions [26].

While FNS I and F3H competes for the same substrates, FNS I catalyzes the desaturation of (2S)-flavonones at the C2/C3 position, resulting in the production of flavones [8]. In the recombinant *OsFNS I-1* reaction system cloning from rice, the OsFNS I-1 activity increases gradually along with the increase in AsA concentrations from 50 μM to 2 mM, indicating that AsA is the key cofactor for OsFNS I-1 [27].

The formation of F3H provides FLS with substrates, which introduces a double bond at the dihydroflavonol’s C2/C3 positions [8]. According to an evaluation of molecular docking, AsA is the best activator for FLS among the possible small molecules such as piperazine-1,4-bis, 2,6-pyridinedicarboxylic dichloride, etc., [28]. In FLS-AsA complex forms, hydrogen bonds formed between the ligand and protein improve the stiffness and stability of the complex [28].

Dihydroflavonol 4-reductase (D4R) catalyzes the formation of F3H, resulting in leucocyanidin formation, which is a precursor used by ANS to produce anthocyanidins [29,30]. The absence of AsA in the VvANS reaction system in vitro enzyme kinetic studies results in the formation of no product, suggesting that AsA is required for *Vitis vinifera* ANS activity [31]. In the absence of AsA, *Arabidopsis thaliana* ANS only exhibited 5% of full activity using standard in vitro assay conditions [32]. The role of AsA is to reduce the reactive oxidizing species produced by the uncoupled turnover of 2-oxoglutarate [33]. AtANS catalyzes the conversion of trans-dihydroquercetin to quercetin when AsA concentrations are high [34]. The homolog of ANS is leucoanthocyanidin dioxygenase (LDOX), and the two work in parallel pathways to generate different proanthocyanins [35]. LDOX shares the same characteristic in that it needs 2-oxoglutarate, Fe (II), and 2-oxoglutarate to exert biochemical activity [35].

### 2.2. AsA Participate in Hydroxylation on O-Glycoproteins and Response to Hypoxia

Plant cell walls are made up of hydroxyproline-rich glycoproteins (HPRGs), which are responsible for the structure, rigidity, dynamic architecture, and sensing function of the cell wall [36]. According to the degree of glycosylation, HPRGs are divided into several subgroups: arabinogalactan proteins (AGPs), extensins (EXTs), and repetitive proline-rich proteins (PRPs) [37]. Prolyl 4-hydroxylase (P4Hs), a class of 2-ODDs family enzyme, catalyzes the hydroxylation of continuous proline residues during HPRG production in plant cell walls; this process is highly dependent on AsA involvement [36]. P4H activity is observed to respond to hypoxia, indicating that AsA is closely associated with changes in anoxia sensitivity [13,38,39].

Different P4Hs react differently to changes in AsA concentration. It has been reported that two P4Hs of *Arabidopsis thaliana*, AtP4H1 and AtP4H2, have the highest *Km* values when AsA is used as a cofactor other than Fe (II) or 2-oxoglutarate [40]. AtP4H5 can hydroxylate the EXT peptide in vitro when 200 μM AsA is present [41,42]. Treatment with α, α-dipyridyl (DP) and ethyl-3,4-dihydroxybenzoate (EDHB) inhibits hydroxylation, increasing *AtP4H5* expression and root hair cell density [41,42,43]. Eleven P4Hs that catalyze the synthesis of AGPs have been identified in *Nicotiana benthamiana* with sequences similar to those found in *Arabidopsis thaliana* [44]. Since AsA is necessary for these enzymes to function, it may be inferred that in the absence of AsA, all P4H is inactive [44]. Furthermore, AsA affects the arrangement of AGPs by minor textural modifications. For example, in tomato, the cell walls become more undulating when treated with 6 mM AsA than during 1.5 mM and 3 mM AsA treatment [45]. Recently, it is discovered that *Arabidopsis thaliana*’s highly AsA-dependent AtP4H3 responds to hypoxia in the cells of the root stele, columella, leaf tips, and stipules [46].

### 2.3. AsA Highly Related to Plant Hormones Anabolic and Catabolic Metabolism

A group of 2-ODDs known as AsA are involved in the metabolism and anabolism of plant hormones, including gibberellins (GA), indole-3-acetic acid (IAA), salicylic acid (SA), and ethylene biosynthesis [8]. As essential signaling molecules, hormones regulate different aspects of plant development and physiology, and thus affect crop yield and the response to abiotic stresses [10,16,19].

GA anabolic and catabolic genes encode 2-ODDs, including gibberellin 3 β-hydroxylase (GA3ox), GA20ox and GA2ox, which catalyze hydroxylation of GA9 and GA20 at C3 position, decarboxylation of GA12 and GA53 at C20 position, and hydroxylation of GA backbone at C2 position, respectively [16,47]. An increase in AsA concentration in cell-free extracts from developing *Cucurbita maxima* embryos stimulates the hydroxylation of 3β to GA13 and 12α to 12α-hydroxyGA25 [48]. The AsA-deficient *vtc1* mutant of *Arabidopsis thaliana* exhibits a reduced amount of bioactive GAs, while exogenous GA3 treatment facilitates the growth of the *vtc1* mutant [49]. These findings support the hypothesis that partially inactive AsA-dependent enzymes involved in GA synthesis are those whose AsA content falls below a certain threshold level [50]. The AsA content of *Eucalyptus globules* increases after treatment with oligo-cattageenan which in turn promotes an 8.2-fold increase in the GA3 level [50]. On the contrary, treatment of *Eucalyptus globules* with lycorine, an inhibitor of AsA synthesis, results in a partial inhibition of GA3 synthesis [50].

One of the most important enzymes in the catabolism of indole-3-acetic acid (IAA) is auxin oxidase 1 (DAO1), a member of the 2-ODDs [8,51]. It catalyzes inactive IAA into 1-oxindole-3-acetic acid (oxIAA) [52]. Recombinant rice DAO protein exhibits the catalytic activity of IAA oxidation in in vitro assays with 5 Mm AsA present [52]. High-throughput analyses of auxin-hungry BY-2 cells reveal a significant downregulation of DAO1 at both the transcriptional and protein levels. Consistently, oxIAA levels are reduced in the following three mutants: *DAO1* siRNA-silencing in tobacco BY-2 cells, CRISPR/Cas9-*NtDAO1* seedling, and *dao1-1* mutant in *Arabidopsis thaliana* [51].

Salicylic acid 3-hydroxylase (S3H) is identified and characterized as a 2-ODDs in the catabolism of salicylic acid (SA), converting SA into 2,3-dihydroxybenzoic acid (2,3-DHBA) and 2,5-DHBA [53]. SA is converted to 2,3-DHBA or 2,5-DHBA conjugates in in vitro characterization systems when Fe (II), AsA, 2-OD, and H_2_O_2_ are present [53].

Ethylene regulates the growth and development of plants and the ripening of fruit. Ethylene biosynthesis begins with L-methionine to S-adenosyl-L-methionine (SAM), while ACC oxidase (ACO), a 2-ODD, catalyzes the conversion of 1-aminocyclopropane-1-carboxylic acid (ACC) to ethylene in the last step [8,54]. Unlike other classical 2-ODDs, ACO does not use 2-oxoglutarate as a substrate [55]. Alternatively, by providing two electrons, AsA mediates the reaction from CO_2_ to bicarbonate, which subsequently oxidizes ACC to produce ethylene [55].

### 2.4. AsA Is Beneficial to DNA Damage Repair

In humans and *Escherichia coli*, the RNA/DNA demethylase ALKB, which is highly dependent on AsA, is responsible for the alkylation of nucleic acids for DNA damage repair [20]. In the case of several different substrates, such as 1-methyladenine, 1-methyl-2′-deoxyadenosine, 3-methylcytidine, and 3-methyl-2′-deoxycytidine, the conversion rate of 2-oxoglutarate is highly correlated with AsA [56]. Treatments with additional reduced compounds such as β-mercaptoethanol, tris phosphine, 4-nitrocatechol, and dithionite would result in the loss of ALKB activity [56]. Currently, very little is known about the demethylation that ALKB-related proteins catalyze in plants. The *E. coli* ALKB homologous protein alkbh2 is identified in *Arabidopsis* [56], which is able to increase the survival rate of the *E. coli alkB* mutant strain to a normal level [20]. However, the impact of AsA on alkbh2 activity needs further investigation.

## 3. As an Antioxidant or Pro-Oxidant, AsA Responds to Oxidative Stress through Electron Recycling

To maintain redox balance, the AsA-GSH cycle may supply electrons for a variety of important physiological and biochemical pathways [3,6,7,57]. In general, AsA participation is necessary for the smooth transportation of photosynthetic electrons during photosynthesis and the consumption of excess energy [58]. Additionally, the AsA-GSH cycle is effective at scavenging reactive oxygen species for heavy metals and oxidative stress in plants [6,7].

### 3.1. AsA Act as an Electron Donor for Photosynthetic Systems

Photosynthesis in plants starts at the thylakoid membrane with two light reactions taking place simultaneously at photosystem I (PSI) and photosystem II (PSII) [57,59] (Figure 3). PSI is a chromoprotein complex composed of multiple protein subunits on the photosynthetic membrane which catalyzes the transfer of electrons from the inside to the outside of the thylakoid membrane [60]. The photosynthetic electrons are transferred from plastocyanin (PC) to reduced ferredoxin (Fd) [59]. PSII is a photosynthetic unit exerting a water/plastoquinone (PQ) oxidoreductase function in the thylakoid membrane, including the peripheral light-harvesting chromophore protein complex (LHCII), the reaction center (RC) P680, and the manganese complex [Mn_4_O_5_Ca] [58].

AsA could supply electrons to PSII in the natural state [61,62]. The half-time of electron donation (t_1/2_) to PSII in *Arabidopsis thaliana* AsA-deficient mutant *(vtc2*) leaves is significantly prolonged, reaching 55 ms, while the wild-type is only 25 ms [61]. Furthermore, exogenous supplementation of AsA can restore the t_1/2_ to PSII in *vtc2* mutants to normal levels [61]. Under acidic pH or UV-B conditions, AsA is protoxidized in PSII to replace water as the electron donor [63]. Additionally, AsA has the ability to donate electrons to PSI in maize bundle sheath cells [64]. The electron contribution rate from AsA to PSI reaches approximately 50–100 μ equivalents (mg Chl)^−1^ at the AsA concentration of 70–80 mM [65]. Under saturated light conditions, leaf CO_2_ uptake is reduced by approximately 50% in two ethyl methane sulfonate-induced AsA synthesis defective mutants (*GGP-5261* and *GGP-49C12*), suggesting that AsA is the main limitation for photosynthesis [66]. As photosynthesis proceeds, ATP generation on the inner membrane of the thylakoid is accompanied by the production of harmful reactive oxygen species (ROS, e.g., hydroxyl radical •OH, hydrogen peroxide H_2_O_2_, superoxide radical O_2_^•−^) [67]. In relation to plant growth, development, and stress responses, AsA has the action of scavenging ROS by donating electrons [68]. For example, the growth of axillary buds is in a quiescent state under high H_2_O_2_ levels in rosebush; in turn, with the rise in AsA content, H_2_O_2_ scavenging is induced, which finally allows axillary buds to outgrow [69].

To cope with variation in light intensity, plants have evolved non-photochemical quenching (NPQ) to dissipate excess absorbed light energy as heat [70]. NPQ requires an active xanthophyll cycle consisting of two de-epoxidate processes, both catalyzed by de-epoxidase (VDE) which are the conversion of violaxanthin to antheraxanthin and the sequential conversion to zeaxanthin (Figure 3) [71]. Numerous studies have shown a strong correlation between the concentration of AsA and the rate of NPQ and VDE [70,71,72]. The NPQ rate of the AsA-deficient mutant *vtc2* is significantly lower than that of the wild-type *Arabidopsis thaliana* after being exposed to 1500 μmol photons m^−2^ s^−1^. Furthermore, violaxanthin to zeaxanthin conversion is slower in *vtc2* mutants than in the wild-type plant [72]. In vtc2 mutants, the NPQ and de-epoxidation rates are restored after 160 min of feeding detached leaves with 10 mM exogenous AsA [72]. The effective dissipation of surplus energy and the indirect elimination of ROS are made possible by the connection between AsA and NPQ and VDE.

### 3.2. AsA Enhances Heavy Metals Tolerance in Plant Cells

Similar to photosynthesis, AsA also has a scavenging effect on harmful ROS induced by heavy metals in plant cells [73]. In fact, the types of heavy metals in soil have a great diversity and are mainly composed of cadmium (Cd), chromium (Cr), Hydrargyrum (Hg), Aluminum (Al), Copper (Cu), and Lead (Pb), etc. All heavy metals are easily absorbed by plant root cells and then transported to shoot or leaf cells resulting in metal toxicity.

Heavy metals can exist as free metals or in complexes with either organic or inorganic materials, interfering with essential metabolic processes during plant growth and development. Interaction between heavy metals and cell components results in the destruction of signaling pathways and cellular macromolecules. For example, under 10 μM Cd treatment, the malondialdehyde (MDA) content is significantly increased by 36.6% and 54.6% in two different wheat cultivars, respectively [74]. At the same time, the negative oxygen ions, hydrogen peroxide, and MDA content increase drastically in soybean under 10 μM Cr treatment [75]. Furthermore, heavy metals inhibit photosynthesis by inducing chloroplast ultrastructural modification, loss of membrane integrity, and interference with electron transport in the photosynthetic system. Under 250 mg/mL Cd treatment, the chloroplast exhibits a dissolved membrane and damaged lamellar structure with a partially disintegrated grained lamellar [76]. Heavy metals are prone to induce cells to produce methylglyoxal (MG), which can react with biological macromolecules such as proteins and nucleic acids to further induce programmed cell death (PCD) because of its extremely high activity. According to fluorescence microscope results, the toxicity of Cr induces cell death in tomato root [77]. Additionally, heavy metals restrain ROS scavenging by means of blocking several antioxidant enzymes and their downstream responses to promote early oxidative stress, resulting in altered antioxidant status.

AsA alleviates metal-induced plant growth stress by the promotion of enzyme activity and postponement of programmed cell death. The length, dry weight, fitness, and tissue density are reduced by 16%, 32%, 13%, and 31% under 25 μM Cr, while 1 mM AsA and 25 μM Cr co-treatment reduce the above indicators by 2%, 7%, 1%, and 6%, respectively, meaning that exogenous AsA has the ability to reverse Cr toxicity in tomato roots [78]. In addition, the addition of AsA stimulates APX and glutathione-S-transferase (GST) activity by 53% and 28%, accompanied by a lower level of cell death and ROS [78]. A significant increase in the AsA content and APX activities is shown in two okra cultivars, as well as two wheat cultivars exposed to Cr toxicity [79,80]. In the meantime, the change in endogenous AsA content is also used to deal with the change in redox status [81]. After combinatorial treatment for 9 days with different concentrations and types of heavy metals, such as 50 μM Zn with 1 μM Cd, or 100 μM Zn with 10 μM Cd, the content of AsA in *Pistia stratiotes* leaves was reduced by 55–96% compared with the control, while the content of DHA increased by 93–133% [82]. Remarkably, AsA alone enhances the antioxidant capacity to counteract Cd pressure rather than the other antioxidants [83]. In summary, AsA showed significant ameliorative potential against heavy metal toxicity in several plants.

### 3.3. AsA Relieves Oxygen Pressure Induced by Environment Changes in Plant Cells

The AsA-GSH cycle’s constituent chemicals and enzymes respond to changes in the redox state of plant cells. The growth status of plant is closely related to changes in redox status induced by abiotic stress which generally destroys plant cell homeostasis. For instance, salt stress is a major environmental stress that limits plant yield and productivity in many crops. Under 120 mM NaCl salinity stress, both the growth and yield of *Silybum marianum* reduce sharply in all genotypes collected from diverse ecozones [84]. Plant chilling injury (CI) is caused by a series of events, starting with a slight increase in cell membrane viscosity, followed by an increase in ROS leading to further membrane dysfunction, protein oxidation, inhibition of enzymatic activity, and DNA and RNA damage [85]. Importantly, changes in the redox state of plants lead to changes in thiol protein (SH-group) patterns, which in turn generate metabolic perturbations [86]. In both *Lycopersicon esculentum* and *Lepidium sativum* treated with Inuloxin A (InA), the fluorescence of the SH-group, labeled with monobromo-bimane (mBBr), is lower than that of the control [86].

AsA-GSH related substances act as an intracellular buffer to maintain redox balance. Compared with the control group, the AsA content in tomato under NaCl stress is decreased by 41.86%, and the activities of APX, GR, GSH, and GSSG are increased by 86.05%, 28.74%, 50.00%, and 87.07%, respectively, while MDHAR and DHAR are decreased by 38.30% and 32.31% [87]. The AsA content is induced to rise during tomato storage in 4 °C, indicating that AsA has a higher scavenging activity to couple with chilling injury [85]. In response to drought stress, the AsA content in the peel of many tomato cultivators is reduced, especially Fragola, Pisanello, Giallo, Pantano, and Pearson [88]. Heterologous overexpression of *AgGMP,* which is a key gene in AsA biosynthesis derived from celery, increases drought resistance in *Arabidopsis thaliana*, demonstrating the role of AsA in regulating drought stress [89]. Strigolactones (SLs) could also regulate tomato salt tolerance by regulating the gene expression of AsA-related enzymes [90]. Collectively, AsA plays an excellent role in nitrogen deficiency, which is tightly related to secondary metabolism as one of the essential macronutrients in plant cells [91]. Under nitrogen deficiency stress, *CsAPX1* expression level is positively correlated with the nitrogen regulatory protein P-II (*CsGLB1*) expression level. The nitrogen response is influenced by the binding locations of *CsAPX1* and *CsGLB1* [91]. Under conditions of sufficient nitrogen, the simultaneous overexpression of *CsAPX1* and *CsGLB1* in an *E. coli* strain shows a better growth than that of the empty vector under nitrogen-sufficient conditions. The high correlation between *CsAPX1* and *CsGLB1* suggests that cytoplasmic AsA could modulate life activities by cooperating with nitrogen regulatory proteins in the tea plant under nitrogen deficiency [91]. The AsA/DHA and GSH/GSSG ratios exhibit a shift towards the oxidized form under an allelochemical InA stress, offering insight into the response pattern of the plant cell redox system [86].

### 3.4. AsA Has Pro-Oxidative Activity under Iron or Copper Overload

It is worth noting that AsA also exhibits a pro-oxidative effect, which is manifested as accelerating the cycle between the oxidation and reduction states of metal ions. In the case of iron overload, AsA combines with iron or iron chelators to form a hybrid complex and reduces ferric to ferrous ions, resulting in the production of large amounts of ROS, especially hydroxyl radicals [92]. Therefore, AsA is often considered as an adjunct to cancer treatment due to its cytotoxicity in cancer cells [93,94]. In the case of copper overload, Cu(II) rapidly triggers AsA oxidation and completes the reaction within 50 min [95]. These studies have demonstrated that the overloading of iron or copper triggers the pro-oxidation of AsA, which is helpful in understanding the pro-oxidative effect of cancer cell removal and its application in the biomedical field [96].

## 4. As a Substance or Precursor, AsA Participates in Metabolism Directly

In addition to serving as a cofactor, AsA can directly participate in physiological and biochemical reactions. First, AsA directly participates in epigenetic reprogramming to regulate DNA demethylation in plant cells. Second, the AsA cleavage product serves as a precursor for the production of organic acids. Finally, AsA and cell wall polysaccharides synthesis share the same pathway.

### 4.1. AsA Directly Participate in Demethylation Modification as an Important Substrate

The methylation status of DNA cytosine is a chemical signal, which is closely correlated with genome stability, gene imprinting, plant development, and the response to the environment without changing the DNA sequence. AsA provides glycerol at C4-C6 for a unique demethylation modification, C5-glyceryl-methylcytosine (5gmC), which is described as a homolog in *Chlamydomonas reinhardtii* for the first time. The modification into 5gmC is catalyzed by the 5mC-modifying enzyme (CMD1) which is a homologous protein of the mammalian ten-eleven translocation enzyme (TET) in demethylation [97]. More importantly, AsA is exclusively demanded for CMD1 catalytic activity other than some additional reductants such as dithiothreitol (DTT), vitamin E, or DHA [97]. In comparison to the wild-type, *Chlamydomonas reinhardtii* exhibits a 60% decrease in 5gmC levels and a doubling in 5mC with CMD1 knockout. In the meantime, the 5gmC levels decrease by 80% and 5mC doubles in the AsA-deficient *vtc2* mutant of *Chlamydomonas reinhardtii* [97].

Recently, the structure of CMD1 and the center of activity for the enzyme have been determined [98]. The catalytic domain of CMD1 contains a typical double-stranded β-helix (DSBH) fold. In the structure of the CMD1-5mC-DNA-AsA complex in vitro, AsA is present in the form of lactone, which does not cause notable conformational changes in the CMD1 active site [98]. A number of direct or indirect hydrogen-bonding and hydrophobic interactions are formed between AsA lactone and CMD1 active residues [97]. The binding of AsA and CMD1, which is essential to the activity of CMD1, is facilitated in large part by the hydrophilic contacts present [98].

### 4.2. AsA Is a Precursor of Organic Acid Biosynthesis

L-threonate, the result of cleavage at the AsA backbone’s C2/C3 location, has the potential to be employed as a synthetic precursor for both oxalic and tartaric acids. Significant increases in the amount of oxalic acid are found in both *Pistia stratiotes* and *Yucca torreyi* when AsA precursors are labeled with the isotope C14 [99,100,101]. In mesophyll oxalate-defective 1 (*mod1*), *mod2*, *mod3*, and *mod4*, the oxalic acid content is restored to wide-type levels by AsA feeding in *Tribulus sativa* [102]. Genes involved in AsA metabolism, especially *MDHAR*, have been thought to play an important role in oxalic acid production, as well as cell expansion and tightening [103].

L-threonate, which is produced during AsA oxidation, could also be decarboxylated to produce L-glycerate or transformed to L-tartrate [104]. The metabolite content analysis shows that the AsA-tartaric acid pathway is more dominant than the glyoxylic acid–oxalic acid fragrance conversion pathway in *Eleusine coracana* [104]. In the non-tartaric acid accumulating species *Ampelopsis aconitifolia*, the AsA level is approximately four times higher than in common berries [105].

### 4.3. AsA Is Related to Cell Wall Metabolism and Fruit Softening

L-galactose-1-phosphate (*GPP3*, also known as *IMP3*) also occupies a prominent position in cell wall synthesis, catalyzing the dephosphorylation of myo-inositol monophosphate to generate free myo-inositol [106]. In the *SlIMP3*-overexpression tomato, not only do AsA and inositol content significantly increase, but the cell wall thickness also increases, accompanied by an increase in fruit hardness, delayed fruit softening, reduced water loss, and extended shelf life. The content of cell wall polysaccharides increases in the *SlIMP3*-overexpression tomato, including uronic acid, rhamnose, xylose, mannose, and galactose in fruit cell walls [107]. The free myo-inositol is catalyzed by inositol oxygenase (MIOX) to produce D-glucuronic acid, which is also a joint substrate for AsA and cell wall synthesis [108]. The AsA content in *MIOX4*-overexpression transgenic plants is higher than that in the control group [108,109,110]. In *Arabidopsis thaliana* with overexpression of MIOX1 and MIOX2, the total monosaccharides originating from the inositol pathway of the cell walls are significantly decreased [111]. A multi-omics analysis of chromosome-level genome assembly in combination with metabolic analysis shows that the expression of the cell wall and starch metabolism genes are highly correlated with the AsA synthesis genes of *Psidium guajava* [112].

Furthermore, changes in AsA content could also affect the composition of cell wall polysaccharides. For example, a comparison between wild-type rice and the *OsVTC1-1* RNAi line shows reduced levels of galactose, mannose, and glucuronic acid in the *OsVTC1-1* RNAi line compared to those in the wild-type seedling [113,114].

## 5. Other Metabolic Pathways Affected by AsA

In addition to reducing in the ATP content, energy charge, ATPase activity, and transcriptional expression of three important enzymes involved in the tricarboxylic acid (TCA) cycle, including citrate synthase (CS), acetyl-CoA carboxylase (ACC), and acetyl-CoA synthetase (ACS), 2% AsA could effectively inhibit the yellowing of fresh-cut yams. In addition, AsA treatment significantly reduces the content of carotenoids, flavonoids, and bisdemethoxycurcumins, and inhibits the enzymatic activity of lycopene β-cyclase (CHYB), β-carotene hydroxylase (LCYB), chalcone synthase (CHS), chalcone isomerase (CHI), diketide-CoA synthase (DCS), and curcumin synthase (CURS), as well as their transcriptional expression levels [115].

Notably, in a recent discovery it was found that exogenous AsA acts as the center of plant signaling transduction and triggers a transient increase in extracellular calcium activity in *Arabidopsis thaliana* roots [111]. In fact, the addition of 40 mM AsA in pipette solutions results in the appearance of a large inwardly directed current, which is three times larger than the current measured using the Gluconate pipette [116].

## 6. Conclusions and Future Prospect

AsA, a type of essential cellular water-soluble vitamin, is involved in vital physiological and biochemical pathways directly or indirectly that regulate plant growth, development, and abiotic stress response stress, in addition to maintaining redox balance. The various physiological and biochemical functions of AsA in plants have been systematically summarized in this review, with a focus on the following: directly participating in metabolism as a precursor or substrate, indirectly functioning in metabolism as a cofactor; and providing electrons to maintain redox balance. This information is crucial for better understanding of the function of AsA in plant cells and to utilizing AsA more efficiently. In conclusion, this review might assist in revealing the metabolic networks AsA is implicated in and provide more clues in enhancing the environmental stress tolerance of plants.

Future research on the function and mechanism of AsA is mainly suggested to focus on the following two aspects. On the one hand, a number of bioinformatics techniques, including multistage virtual screening, molecular docking, and molecular dynamic stimulation, are needed to explore the binding mechanism of AsA and 2-ODDs. In higher plants, however, AsA has a considerable influence on the bifunctional demethylase/glycosylases ROS1 and intracellular 5mC levels in higher plants, which should be explored.

## Figures and Tables

**Figure 1 ijms-25-01832-f001:**
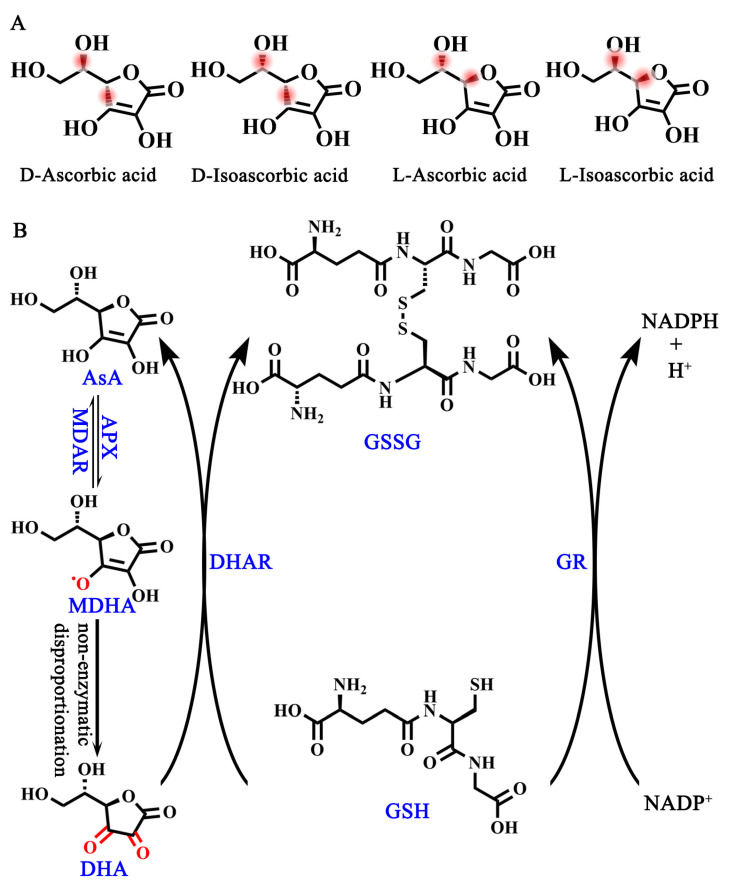
AsA isomers and AsA-DHA cycle. (**A**) Four isomers of AsA. (**B**) AsA-GSH cycle. MDAR, monodehydroascorbate reductase; APX, ascorbate peroxidase; DHAR, dehydroascorbate reductase; GR, glutathione reductase.

**Figure 2 ijms-25-01832-f002:**
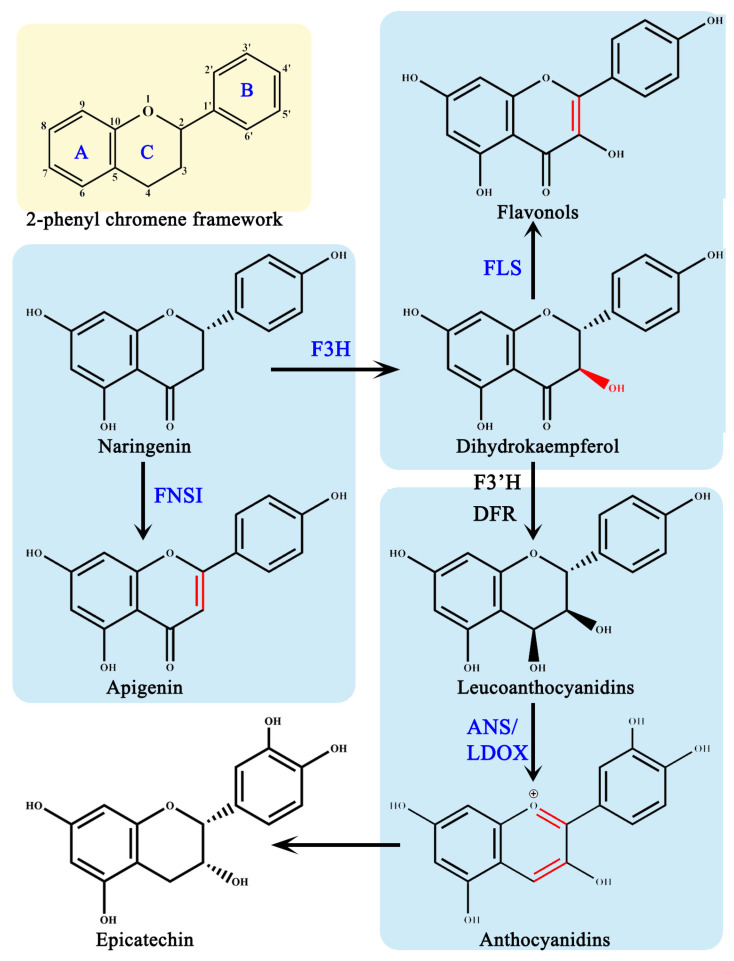
2-dioxygenase in the flavonoid synthesis pathway. F3H, flavonone 3β-hydroxylase; FNS I, flavones synthase I; FLS, flavonol synthase; ANS, anthocyanidin synthase; LDOX, leucoanthocyanidin dioxygenase; DFR, dihydroflavonol 4-reductase; F3′H, flavonone 3′-hydroxylase; A-C, three different rings on the 2-phenyl chromene framework.

**Figure 3 ijms-25-01832-f003:**
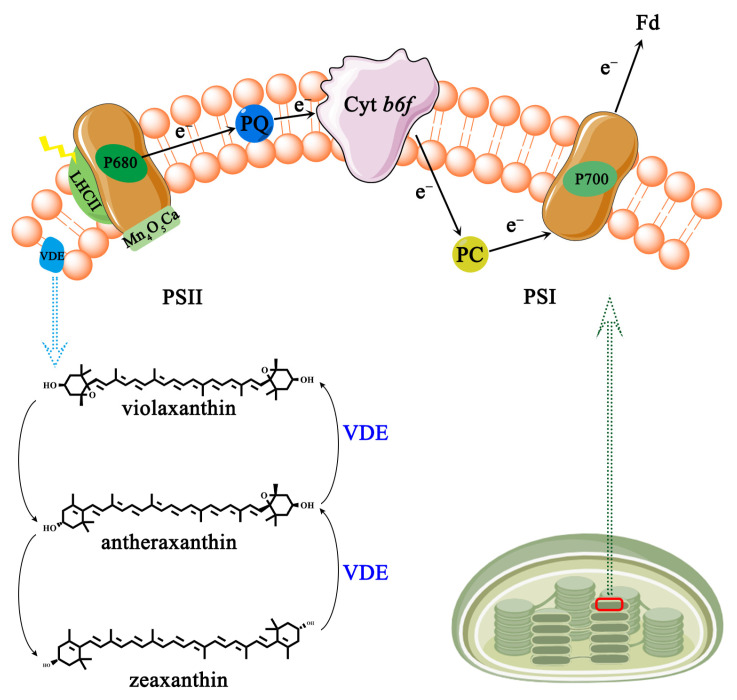
Photosynthesis electron transport chain and vioxanthin cycle. PSI, photosystem I; PSII, photosystem II; PC, plastocyanin; Fd, ferredoxin (Fd); PQ, water/plastoquinone; LHCII, chromophore protein complex; P680 and P700, two reaction centers; VDE, de-epoxidase.

**Table 1 ijms-25-01832-t001:** 2-oxoglutarate/Fe (II)-dependent dioxygenase in plants.

Function	Biochemical Pathway	Enzyme	Substrate	References
Hydroxylation and desaturation	Flavonoid synthesis	F3H	(2S)-flavonones	[18]
		FNS I	(2S)-flavonones	
		FLS	Dihydroflavonols	
		ANS/LDOX	anthocyanins	
Hydroxylation	O-glycosylation modifications HIF-hydroxylation	P4H	proline-rich glycoproteins	[11]
Hydroxylation and decarboxylation	GA anabolism and catabolism	GA3ox	GA9 and GA20	[19]
		GA20ox	GA12 and GA53	
		GA2ox	GA19, GA44 and GA53	
Desaturation	IAA catabolism	DAO	IAA	[19]
Hydroxylation	SA catabolism	S3H	SA	[19]
Desaturation	Ethylene biosynthesis	ACCO	1-aminocyclopropanecarboxylic acid	[19]
Demethylation	DNA repair	Alkbh2	1-methyladenine	[20]

## Data Availability

The original contributions presented in the study are included in the article; further inquiries can be directed to the corresponding author.

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
