# Peer review of "Multiple Physiological and Biochemical Functions of Ascorbic Acid in Plant Growth, Development, and Abiotic Stress Response"

_ijms, 2024, doi:10.3390/ijms25031832_

Round 1
Reviewer 1 Report
Comments and Suggestions for Authors
Manuscript ID ijms-2781633-peer-review
Multiple physiological and biochemical functions of ascorbic acid in plants growth, development and abiotic stress response.
Peiwen Wu, Bowen Li, Ye Liu, Zheng Bian, Jiaxin Xiong, Yunxiang Wang, Benzhong Zhu
The authors showed the structure of ascorbic acid (AsA) and its isomers.
They presented the scheme of biosynthesis of the studied compound, dwelt on three possible branches of its biosynthesis pathway. The functions of ascorbic acid were very evidently presented. First of all, they showed its participation as a cofactor of enzymes responsible for the transformation of phytohormones (gibberellins, salicylic acid, PPIs and ethylene), its participation in the synthesis of various secondary compounds (flavonoids and glucosinolates). Thus they showed the possibility of ascorbic acid participation in plant growth and development. Interesting information on the function of AsA in enhancing stress tolerance in plants is presented, which is examined on the example of hypoxia, salinity, low positive temperatures and heavy metals. The reduction of Fe(III) to Fe(II) for catalyzed 2-oxoglutarate/Fe(II)-dependent center dioxygenase is the mechanism of AsA function under hypoxia. AsA has an antioxidant function. The AsA-glutathione cycle is the most sought after in the plant. The same cycle can participate as an electron donor for photosynthetic systems, to maintain redox balance and response to abiotic stress in plants.
AsA is directly involved in metabolism, as a substrate. It regulates DNA demethylation, and the cleavage product of AsA is a precursor to the biosynthesis of organic acids (e.g. oxalic acid and tartaric acid). AsA is associated with cell wall metabolism and fruit softening. AsA acts as a plant signal transduction center and causes a temporary increase in extracellular calcium activity in roots.
Remarks:
1. The review is silent on the prooxidant functions of AsA. AsA as a possible pro-oxidant should be discussed.
2. You should add "pro-oxidant" to the Keywords and Abstract (Lines 22 and 19).
3. You should remove "multiple functions" from the Keywords (Line 22).
4. You should add the title of the article to Reference 12 (Line 478).

Author Response
Response to Reviewer 1 Comments
1. Summary
Thank you very much for taking the time to review this manuscript. Please find the detailed responses below and the corresponding revisions highlighted in the re-submitted files.
2. Point-by-point response to Comments and Suggestions for Authors
Comments 1: The review is silent on the prooxidant functions of AsA. AsA as a possible pro-oxidant should be discussed.
Response 1: So much appreciation to the reviewer with good suggestions for improving our manuscript quality. We have added more details to describe the pro-oxidant effect of AsA as red-marked in Line 328-338.
Comments 2: You should add "pro-oxidant" to the Keywords and Abstract (Lines 22 and 19)
Response 2: Thanks a lot. We have added “pro-oxidant” to the Abstract and Keywords in Lines 19 and 22 which are marked as red in manuscript.
Comments 3: You should remove "multiple functions" from the Keywords (Line 22).
Response 3: OK. We have removed “multiple functions” from the Keywords in Line 22.
Comments 4: You should add the title of the article to Reference 12 (Line 478).
Response 4: Thank you. We have added title of the article to Reference 12 (Line 490).
Reviewer 2 Report
Comments and Suggestions for Authors
Review by Peiwen Wu et al. “Multiple physiological and biochemical functions of ascorbic acid in plants growth, development and abiotic stress response” examines various aspects of the functioning of ascorbic acid in plants.
The review looked at all aspects of ascorbic acid in some detail. The list of references given corresponds to the material and is quite modern.
The structural aspects of the biosynthetic chains are depicted very well with proper figures. The review concerns specifically the physiological functions of ascorbic acid in plant metabolism. And this is reflected precisely in the title of the review. This material is worthy of the highest praise.
However, in my opinion, the most important characteristic of ascorbic acid as vitamin C and its role in human nutrition have been omitted. Let me make a reservation right away that the title of the review does not provide for a discussion of such a property, as the title suggests. However, the mention of experiments with overexpression of genes involved in the biosynthesis of ascorbic acid in sections 3 and 4 in transgenic plants could be separated into a separate section with a code name, for example, bioengineering of ascorbic acid. And it would be possible to focus on creating transgenic plants with a higher content of ascorbic acid specifically for human nutrition. However, we leave this issue to the discretion of the authors.
Overall, the review is very good and certainly worthy of acceptance for publication.
​
Author Response
Response to Reviewer 2 Comments
1. Point-by-point response to Comments and Suggestions for Authors
Comments 1: However, in my opinion, the most important characteristic of ascorbic acid as vitamin C and its role in human nutrition have been omitted. Let me make a reservation right away that the title of the review does not provide for a discussion of such a property, as the title suggests.
Response 1: Thank you for putting forward the useful issues regarding the characteristic of AsA and its role in human nutrition. These issues are briefly addressed in the Introduction section of our review. Given that the topic of this review focuses on the physiological function of AsA in plant cells, we do not discuss the above issues in detail. We will prepare a special manuscripte potentially to these issues in the future.
Comments 2: However, the mention of experiments with overexpression of genes involved in the biosynthesis of ascorbic acid in sections 3 and 4 in transgenic plants could be separated into a separate section with a code name, for example, bioengineering of ascorbic acid. And it would be possible to focus on creating transgenic plants with a higher content of ascorbic acid specifically for human nutrition. However, we leave this issue to the discretion of the authors.
Response 2: Much appreciate for your comment and suggestion about bioengineering of AsA as a separate section. This manuscript mainly focuses on the effects of changes in AsA content on plant physiology. And some specific reviews have been already reported about bioengineering of AsA, such as http://dx.doi.org/10.1016/j.copbio.2017.01.011, https://doi.org/10.3389/fpls.2018.02006, etc. So, we will pay more attention to this issue in the future works.
Reviewer 3 Report
Comments and Suggestions for Authors
Dear,
Upon reading your manuascript here are my recomendations:
Figure 1 should be of better quality and the names a little bigger, now it is blurry and hard to read.
Line 89-90: According to the modification of hydroxyl groups on the three rings of basic skeleton C6-C3-C6, flavonoids can be divided into flavanonol, flavone, kaempferol and anthocyanindins.
Please check this information, flavanoids are not devided into these groups, this is not correct! See example, or any other studies in flavonoids! https://doi.org/10.3390/biology11020342
Please correct throught the manuascript
Other than that i have no additional comments
Author Response
Response to Reviewer 3 Comments
1. Summary
Thank you very much for your useful comments. We have read comments carefully and answered below all of reviewers’ questions point by point. The corresponding revised portion are red-marked in the present manuscript.
2. Point-by-point response to Comments and Suggestions for Authors
Comments 1: Figure 1 should be of better quality and the names a little bigger, now it is blurry and hard to read.
Response 1: Thanks for your constructive suggestions and we apologized for not clearly in Figure 1. We have increased the image resolution and enlarged the names to make the picture clear.
Comments 2: Line 89-90: According to the modification of hydroxyl groups on the three rings of basic skeleton C6-C3-C6, flavonoids can be divided into flavanonol, flavone, kaempferol and anthocyanindins.
Please check this information, flavanoids are not devided into these groups, this is not correct! See example, or any other studies in flavonoids! https://doi.org/10.3390/biology11020342
Response 2: The comment is very worthful for improving the quality of the paper. We have carefully read the literature you provided and revised this information at Line 89-91.
“According to the chemical structures, flavonoids have been divided into six subclasses, including flavan-3-ols, flavanones, flavones, flavonols, isoflavones, and anthocyanins.”